# Impact of Anemia Severity on the Outcome of an Aneurysmal Subarachnoid Hemorrhage

**DOI:** 10.3390/jcm11216258

**Published:** 2022-10-24

**Authors:** Maryam Said, Thiemo Florin Dinger, Meltem Gümüs, Laurèl Rauschenbach, Mehdi Chihi, Jan Rodemerk, Veronika Lenz, Marvin Darkwah Oppong, Anne-Kathrin Uerschels, Philipp Dammann, Karsten Henning Wrede, Ulrich Sure, Ramazan Jabbarli

**Affiliations:** 1Department of Neurosurgery and Spine Surgery, University Hospital of Essen, 45147 Essen, Germany; 2Center for Translational Neuro- and Behavioral Sciences (C-TNBS), University Duisburg Essen, 47147 Duisburg, Germany; 3Institute of Transfusion Medicine, University Hospital of Essen, 45147 Essen, Germany

**Keywords:** subarachnoid hemorrhage, anemia, outcome, risk factors

## Abstract

Objective: Previous reports indicate a negative impact of anemia on the outcome of an aneurysmal subarachnoid hemorrhage (SAH). We aimed to identify the outcome-relevant severity of post-SAH anemia. Methods: SAH cases treated at our institution between 01/2005 and 06/2016 were included (*n* = 640). The onset, duration, and severity (nadir hemoglobin (nHB) level) of anemia during the initial hospital stay were recorded. Study endpoints were new cerebral infarctions, a poor outcome six months post-SAH (modified Rankin scale > 3), and in-hospital mortality. To assess independent associations with the study endpoints, different multivariable regression models were performed, adjusted for relevant patient and baseline SAH characteristics as well as anemia-associated clinical events during the SAH. Results: The rates of anemia were 83.3%, 67.7%, 40.0%, 15.9%, and 4.5% for an nHB < 11 g/dL, < 10 g/dL, < 9 g/dL, < 8 g/dL, and < 7 g/dL, respectively. The higher the anemia severity, the later was the onset (post-SAH days 2, 4, 5.4, 7.6 and 8, *p* < 0.0001) and the shorter the duration (8 days, 6 days, 4 days, 3 days, and 2 days, *p* < 0.0001) of anemia. In the final multivariable analysis, only an nHB < 9 g/dL was independently associated with all study endpoints: adjusted odds ratio 1.7/3.22/2.44 for cerebral infarctions/in-hospital mortality/poor outcome. The timing (post-SAH day 3.9 vs. 6, *p* = 0.001) and duration (3 vs. 5 days, *p* = 0.041) of anemia with an nHB < 9 g/dL showed inverse associations with the risk of in-hospital mortality, but not with other study endpoints. Conclusions: Anemia is very common in SAH patients affecting four of five individuals during their hospital stay. An nHB decline to < 9 g/dL was strongly associated with all study endpoints, independent of baseline characteristics and SAH-related clinical events. Our data encourage further prospective evaluations of the value of different transfusion strategies in the functional outcomes of SAH patients.

## 1. Introduction

Anemia is a common condition after an acute aneurysmal subarachnoid hemorrhage (SAH). Although often based on small case series, existing reports indicate a negative impact of anemia on the outcome in SAH patients [1,2,3].

A decline in hemoglobin (Hb) value during a SAH has repeatedly been described, often in the context of aneurysm surgery and conservative management of delayed cerebral ischemia [4]. Associations between Hb decline and poorer clinical condition, longer hospital stay, unfavorable outcome, and increased mortality have been reported [5,6]. Despite these findings, there are no specific guidelines or recommendations for managing anemia during a SAH. In fact, in the current literature, the cut-off Hb value for anemia, and with that the threshold for its treatment, varies greatly [7,8]. In addition, it remains unclear whether anemia has an independent impact on SAH outcome or is the consequence of poor initial clinical condition or specific complications during a SAH, such as aneurysm rebleeding, cerebral vasospasm, or systemic infections. Finally, insufficient and inconsistent data are available on the significance of the onset and duration of anemia in the context of its clinical impact.

In our large monocentric observational cohort study, we aimed to analyze the occurrence, time trends, and clinical impact of post-SAH anemia, with a particular emphasis on identifying outcome-relevant anemia severity.

## 2. Materials and Methods

### 2.1. Patient Population

For this retrospective study, all patients aged 18 years and older treated for an aneurysmal SAH at our institution between January 2005 and June 2016 were considered eligible. Exclusion criteria for this study were: (1) no treatment for the SAH was received, (2) late admission (> 48 h after ictus), and (3) pre-existing anemia in the context of another disease. The study was registered in the German trial registry (DRKS, Unique identifier: RKS00008749) and approved by the local ethics committee (Ethik-Kommission, Medizinische Fakultät der Universität Duisburg-Essen, Registration number: 15-6331-BO).

### 2.2. SAH Management

The bleeding source was diagnosed using digital subtraction angiography (DSA) in all patients who presented at our institution with a suspected SAH. After a ruptured intracranial aneurysm was confirmed, microsurgical clipping or endovascular coiling and/or stenting was commonly performed within 24 h after admission.

Conservative management included oral nimodipine for 21 days and the maintenance of normovolemia and mean arterial pressure > 70 mmHg. Vasospasm surveillance consisted of daily neurological assessment and transcranial Doppler ultrasound (TCD). In addition, patients underwent repeated DSAs for the identification and, if confirmed, endovascular treatment of vasospasm [9] in the following cases: (1) neurological worsening, defined as a new neurological deficit or a ≥ 2 points decline in the Glasgow coma scale not attributable to other complications (such as rebleeding or hydrocephalus); (2) in unconscious patients, development of absolute mean flow velocities > 120 cm/s or an increase of more than 50% compared to the previous measurement in TCD. In the SAH patients with clinical and/or angiographic signs of cerebral vasospasm, conservative management was escalated by increasing the target mean arterial pressure to > 90 mmHg.

Acute hydrocephalus was treated with an external ventricular drain. This device also allowed for continuous intracranial pressure (ICP) monitoring. In cases of pathologically increased ICP (> 20 mmHg) refractory to conservative management [10], decompressive craniectomy (DC) was performed.

Regular laboratory measurements, performed three times weekly for at least 14 days after ictus, assessed anemia occurrence during the SAH, with an increased frequency of blood sampling if clinically indicated. According to the institutional standards based on the widely accepted recommendation for critical care patients [11], red blood cell transfusion (RBCT) was performed in case of a decrease in nadir Hb (nHb) to < 7.0 g/dL.

Computed tomography (CT) scans were performed at admission, within 24 h after aneurysm treatment (or any other neurosurgical procedure), in case of neurological deterioration, and during weaning from external ventricular drainage. There were no significant changes in the institutional SAH management protocol during the reported years.

### 2.3. Data Management

Anemia severity was assessed according to the nHb levels which were documented during the 14 days after the SAH at predefined cutoffs: < 11 g/dL, < 10 g/dL, < 9 g/dL, < 8 g/dL, and < 7 g/dL. The onset (since ictus, in days) and duration (number of days with each nHb level) of anemia were also documented. Furthermore, data on patients’ demographic characteristics, previous comorbidities and regular medication, initial SAH severity, and treatment modality, and on the occurrence of certain adverse events, were collected from the institutional retrospective aneurysm database. The patients’ initial clinical condition was recorded using the World Federation of Neurosurgical Societies (WFNS) scale [12], with further dichotomization into good (WFNS = 1–3) and poor (WFNS = 4–5) grades. Furthermore, radiographic severity was graded according to the original Fisher scale [13], which was also dichotomized (grade 3–4 vs. 1–2) for further analysis. The following complications during the SAH were recorded: aneurysm rebleeding, cerebral vasospasm (based on the presence of the above-mentioned neurologic and/or angiographic characteristics), ICP increase necessitating DC, acute coronary syndrome (ACS), and systemic infections. Finally, functional outcome was assessed at discharge and 6 months after the SAH using the modified Rankin scale (mRS) [14], with an mRS > 3 regarded as an unfavorable outcome.

### 2.4. Study Endpoints and Statistical Analyses

As the primary study endpoints, we investigated the impact of post-SAH anemia severity on the occurrence of the following major outcome events: (1) occurrence of cerebral infarction(s); (2) in-hospital mortality; (3) unfavorable outcome at 6 months post-SAH. The associations between each nHb level and all three outcome events were tested using a univariate analysis and two models of a multivariable binary logistic regression analysis. The univariate analysis was based on the chi-squared or Fisher’s exact test, as appropriate. In the multivariable analysis, the first model (M1) included the following confounders: patients’ age (dichotomized at the cohort’s mean age), sex, WFNS and Fisher grades at admission, presence of acute hydrocephalus, and length of hospital stay. Along with all covariates from the M1 model, the second model (M2) of the multivariable analysis was enhanced by adverse events during the SAH which were significantly associated with the occurrence of anemia and the primary outcome endpoints.

Furthermore, the following analyses were also performed as the secondary study endpoints: (a) the relationship between anemia severity with its onset and duration; (b) the impact of the onset and/or duration of the outcome-relevant anemia severity on the above-mentioned major outcome events; (c) the associations between patients’ baseline and SAH characteristics with the severity of post-SAH anemia. For normally distributed continuous variables, analyses were performed with the Student’s *t*-test. Non-normally distributed continuous variables were analyzed with the Mann-Whitney U-test and one-way ANOVA, as required. The missing values were replaced by multiple imputations. Data analysis was performed using SPSS statistical software (version 27.0). Correlations with a *p*-value of < 0.05 were considered statistically significant.

### 2.5. Data Availability Statement

Any data not published within the article will be shared in an anonymized manner on request with any qualified investigator.

## 3. Results

After excluding non-eligible cases (no aneurysm treatment: *n* = 33; late admission: *n* = 96; pre-existing anemia: *n* = 9), a total of 640 patients were included in the final analyses. The baseline characteristics of the cohort are presented in Table 1.

### 3.1. Post-SAH Anemia: The Prevalence and Timing

There was a decreasing rate of anemia in the cohort, depending on its severity. The lower the documented nHb value during the first two weeks of SAH treatment, the less frequent was the rate of anemia severity (Figure 1). In particular, a decrease of nHb to < 11.0 g/dL was very common in our SAH cohort, affecting a little more than four out of five SAH patients. On the other hand, the most severe anemia with an nHb < 7.0 g/dL was a sporadic condition observed in < 5% of the cohort.

There was also a significant association between the onset of the anemia and its severity (Figure 2). With increasing anemia severity, a later onset in the course of the SAH was observed: on post-SAH days 2, 4, 5.4, 7.6, and 8 for an nHb < 11.0 g/dL, < 10.0 g/dL, < 9.0 g/dL, < 8.0 g/dL, and < 7.0 g/dL, respectively (*p* < 0.0001). Of notice, the number of days patients suffered from anemia was also significantly dependent on its severity. On average, an nHb < 7.0 g/dL lasted for no more than 2 days, whereas an nHb < 11.0 g/dL was seen for considerably longer in patients (mean: 8 days, 6 days, 4 days, 3 days, and 2 days for an nHb < 11.0 g/dL, < 10.0 g/dL, < 9.0 g/dL, < 8.0 g/dL, and < 7.0 g/dL, respectively, *p* < 0.0001).

### 3.2. Prognostic Factors for Post-SAH Anemia

Table 2 presents the summary of the odds ratios for the associations between patients’ baseline and initial SAH characteristics along with the occurrence of post-SAH anemia at different nHb cut-offs (see Appendix A for more detailed data on the performed univariate analysis). Patients aged 55 years and older had a significantly higher risk of developing anemia (nHb < 8.0 g/dL, < 9.0 g/dL, and < 10.0 g/dL) during the SAH. Female sex was also significantly correlated with an nHb < 10.0 g/dL and < 11.0 g/dL. Regarding previous medical history, arterial hypertension (nHb < 8 g/dL and < 9.0 g/dL), hypercholesterinemia (nHb < 10.0 g/dL), diabetes mellitus (nHb < 10.0 g/dL and < 11.0 g/dL), and treatment with statins (nHb < 10.0 g/dL) increased the probability of anemia at different severity levels. Furthermore, the initial severity of the SAH (high WFNS and Fisher scales and the presence of acute hydrocephalus) showed robust associations with the risk of anemia during the SAH, with significance observed for a wide range of documented nHb values (between < 8.0 g/dL and 11.0 g/dL). Finally, the following clinical events during the SAH were also linked with the occurrence of anemia at different severity levels: surgical treatment with clipping and/or DC (for an nHb < 8.0 g/dL, < 9.0 g/dL, < 10.0 g/dL, and < 11.0 g/dL), cerebral vasospasm (nHb < 9.0 g/dL, < 10.0 g/dL, and < 11.0 g/dL), ACS (nHb < 8.0 g/dL), and systemic infections (nHb between < 8.0 g/dL and < 11.0 g/dL). Interestingly, aneurysm rebleeding before treatment did not impact the probability of post-SAH anemia.

### 3.3. Severity of Anemia in Relation to Primary Study Endpoints

The more severe the post-SAH anemia was, the higher was the burden of cerebral infarctions, in-hospital mortality, and unfavorable outcome at six months after the SAH (Figure 3). Univariate analyses showed significant correlations for almost all anemia levels with the analyzed outcome events (see Table 3 with summary data on univariate and multivariate analyses; see Appendix A with detailed analysis of the study endpoints). However, in the final multivariable analysis (M2 model), only an nHb < 9.0 g/dL (adjusted odds ratio (aOR): 1.70, 95% confidence interval (CI): 1.16–2.50, *p* = 0.007) and an nHb < 10.0 g/dL (aOR: 1.83, 95% CI: 1.19–2.81, *p* = 0.006) were independently associated with the risk of cerebral infarction. In both multivariable models (M1 and M2), shorter survival during hospitalization showed a strong link with post-SAH anemia for nHb levels between < 8.0 g/dL and < 11.0 g/dL. Finally, unfavorable outcome at six months was independently associated with an nHb < 8.0 g/dL (aOR: 2.21, 95% CI: 1.21–4.04, *p* = 0.01) and < 9.0 g/dL (aOR: 2.44, 95% CI: 1.57–3.79, *p* < 0.0001) in the enhanced multivariable analysis (M2 model). In summary, post-SAH anemia at < 9.0 g/dL was the only nHb value that showed robust and strong associations with all study endpoints, independently of patients’ outcome- and anemia-relevant baseline parameters, initial SAH characteristics, and clinical events during treatment.

When analyzing the value of the occurrence of an nHb < 9.0 g/dL for patients’ outcome, the onset (post-SAH day 3.9 vs. 6, *p* = 0.001) and duration (3 days vs. 5 days, *p* = 0.041) of anemia showed inverse associations with the risk of in-hospital mortality, but not with the other study endpoints (see Appendix A).

## 4. Discussion

Anemia is a common finding in SAH patients. In our large observational study, we found that four out of five SAH patients suffered from anemia during their hospitalization. In previous literature, several degrees of anemia have been mentioned in relation to patients with SAH. Rates from 30% to almost 60% have been reported [15,16,17,18]. In our study, we found cerebral infarctions, in-hospital mortality, and poor neurological outcome to be independently associated with several degrees of anemia.

Although often described in this patient population, until now no common management strategies have been determined. The use of RBCT in this population is rather restrictive in most institutions [7,19,20,21,22,23]. A North American survey on RBCT practices found that the thresholds for transfusion vary between 7.0 g/dL and 10.0 g/dL [24]. Which RBCT threshold has the best risk–benefit profile for SAH patients is yet to be clarified, as RBCT has been associated with higher rates of thromboembolic events [23], poor outcome [25], vasospasm [26], and cognitive impairment [27].

To our knowledge, our study is the first to describe the occurrence, onset, and duration of a wide range of nHb levels within one large representative SAH cohort. The more severe the anemia, the later the onset in the course of disease. This association reflects the effect of the duration of neurocritical care treatment on the probability and severity of anemia. Therefore, duration of hospital stay is an essential parameter that should be considered in analyses evaluating the association between anemia and SAH outcome. In contrast to our study, where the length of hospitalization was incorporated in the statistical constructs, many previous studies failed to show robust associations between post-SAH anemia and in-hospital mortality [4,19,22,26,28].

Moreover, we were able to show that the more aggravating the anemia, the shorter its duration. This might be the consequence of both later anemia onset and the treating physicians’ lower tolerance of more severe anemia grades. Only an nHb decrease to < 7.0 g/dL was an absolute indication for RBCT in our clinic. Accordingly, with an average of two days, nHb levels < 7.0 g/dL had the shortest duration in the cohort (compared to higher nHb values). Therefore, RBCT in SAH patients with nHb levels < 7.0 g/dL apparently resulted in swifter anemia correction than in SAH patients with less severe nHb levels. This circumstance might partially explain the less prominent link between an nHb < 7.0 g/dL and the study endpoints as compared to higher nHb levels which were tolerated longer than the standard institutional RBCT threshold of < 7.0 g/dL. Another possible explanation for the lack of significance of an nHb < 7.0 g/dL might be the small size of the subpopulation with this nHb, as only 4.5% of our cohort developed such a low nHb.

In our analysis, several baseline characteristics at admission were associated with the development of anemia in the course of the disease. Knowledge of these easily accessible parameters allows for early risk stratification and risk-adjusted anemia management, since many identified anemia predictors (such as patients’ age and initial SAH severity) are also outcome-relevant prognostic factors for SAH patients [29]. Moreover, we found several premorbid conditions to be risk factors for developing post-SAH anemia. In particular, hypercholesterinemia (and statin treatment) and diabetes mellitus were associated with mild forms of anemia (< 10.0 g/dL and < 11.0 g/dL), and the presence of arterial hypertension was associated with an nHb decrease to < 8.0 g/dL and < 9.0 g/dL. There is no literature available on hypercholesterolemia and anemia in SAH patients, but a link between hypercholesterolemia and different chronic types of anemia has previously been described [30]. Patients with essential hypertension [31], and particularly those with pulmonary arterial hypertension [32], were reported to be at higher risk of anemia. Finally, diabetes-related chronic hyperglycemia can also lead to anemia via impaired erythropoietin production [33].

Even if the patient’s previous medical history contributes to anemia risk to some extent, substantial post-SAH anemia is more likely triggered by several factors occurring during the course of the SAH disease. Along with common anemia confounders during intensive care treatment, such as phlebotomy and hemodilution [1,34], there are also SAH-specific clinical events which strongly increase the risk of acute anemia. In particular, surgical treatments, including aneurysm clipping and DC due to refractory ICP increase were significantly associated with anemia risk in our cohort. Both surgical factors had already been reported as anemia triggers in SAH patients [4,7,17]. Moreover, cerebral vasospasms and systemic infections also showed robust correlations with post-SAH anemia. The link between vasospasm and anemia was repeatedly reported in the context of SAH disease. However, the cause–effect relationship between these two events remains unclear; is the anemia the consequence of conservative vasospasm management [35], or should anemia rather be regarded as a marker of poor hemodynamic support, contributing to cerebral hypoxia and resulting in the clinical manifestation of cerebral vasospasm [36]? Or, alternatively, might these two SAH complications exist in a vicious circle? Finally, infections had also previously been reported as risk factors for the development of anemia in SAH patients [16,20]. In the context of intensive care unit treatment, systemic inflammation reduces red blood cell development by inhibiting erythropoietin synthesis, and it interferes with the ability of erythroblasts to incorporate iron [37,38].

A link between anemia and SAH outcome had already been shown in many previous studies [1,3,4,8,28,34,39]. However, these studies suffered from significant limitations with regard to different cut-off values for nHb and the uncertainty of whether anemia was independently related to SAH outcome or merely presented an epiphenomenon of the initial SAH severity and/or of clinical complications during SAH. In our analysis, we adjusted for patients’ baseline characteristics, initial SAH severity, and outcome/anemia-relevant clinical events. We analyzed different nHb levels and found robust independent correlations between an nHb decrease to < 9.0 g/dL and the occurrence of cerebral infarctions, in-hospital mortality, and unfavorable outcome at six months. With the two MVA models we conducted for these analyses, we were able to correct for the hemodiluting effects of SAH therapy. This means that anemia is not in fact an epiphenomenon, but rather an independent predictor in SAH.

Our findings imply a higher RBCT threshold for SAH patients than is currently being practiced in our and many other neurovascular centers, where patients receive transfusions only in cases in which nHb decrease is < 7.0 g/dL. This restrictive transfusion policy is based on the largest to-date prospective clinical trial that evaluated the risks and benefits of RBCT in patients in the intensive care unit [11]. Of note, patients with a SAH were underrepresented in this study, published in 1999. This circumstance, along with the current data on the association between anemia and SAH outcome, necessitates the reconsideration of the current standards for RBCT in SAH patients. Therefore, we recommend a prospective evaluation of the value of different transfusion strategies in the functional outcomes of SAH patients.

## 5. Limitations

Our study has certain limitations, mainly its single-center and retrospective design. Moreover, there is an additional risk of selection and information bias, since the patients with a more aggravating course of illness tend to undergo more invasive procedures and have relatively more blood drawn for tests. Finally, our center does not use invasive monitoring of cerebral oxygenation. These data on the oxygen delivery in the effector tissue would be relevant for confirming the findings and hypotheses of this study. Nevertheless, the present study is based on a large representative SAH cohort. We utilized an extensive explorative analysis, addressing different nHb values with the adjustment of study results for the most relevant baseline patient and clinical characteristics and adverse events during the SAH.

## 6. Conclusions

We found anemia to be very common in SAH patients, affecting four in five individuals during their hospital stay. Interestingly, a decline in nHb to below 9.0 g/dL was significantly related to cerebral infarctions, in-hospital mortality, and poor outcome. These findings are independent of baseline characteristics and adverse clinical events during hospitalization. Previous literature often described anemia as an epiphenomenon of the hemodiluting effects of SAH therapy. Our study provides, for the first time, data on the influence of anemia on poor outcomes in SAH patients, independent of other factors. Our results encourage further prospective evaluations of the value of different transfusion strategies in the functional outcomes of SAH patients.

## Figures and Tables

**Figure 1 jcm-11-06258-f001:**
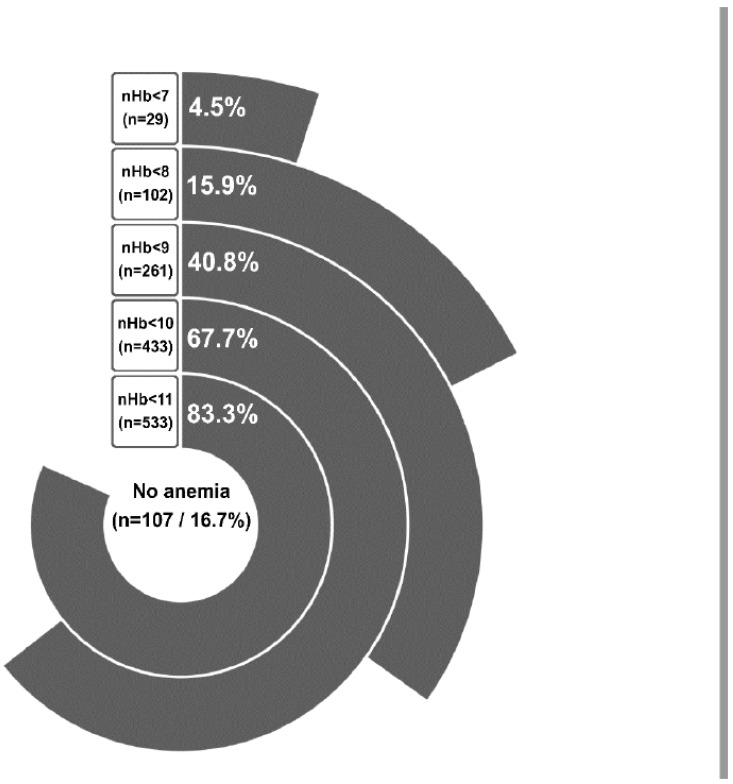
All patients in the cohort are represented based on the nadir Hb (nHb) level. For every anemia subgroup the number of patients and percentage of the total population are depicted, respectively.

**Figure 2 jcm-11-06258-f002:**
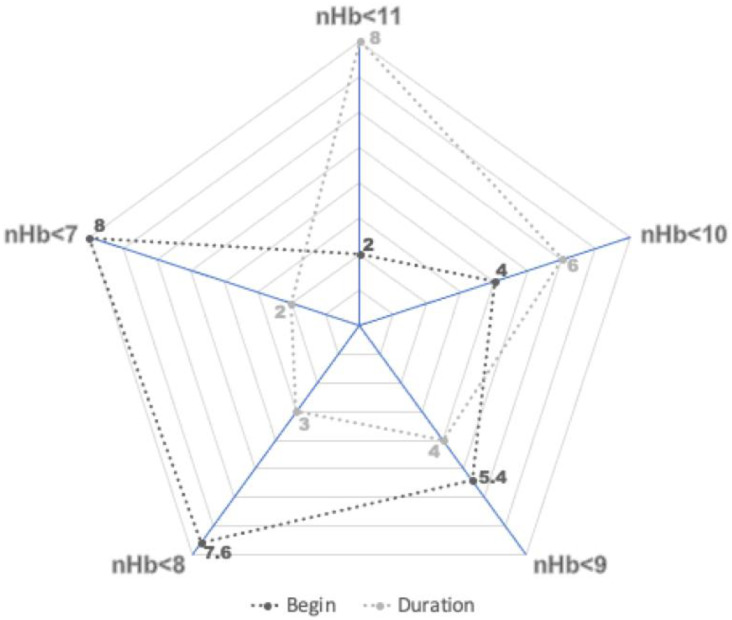
Onset and duration of anemia in days for each subgroup of anemia. For every g/dL drop in nadir hemoglobin (nHb) level, a later onset during the SAH was observed. The duration of the anemia was inversely related to its severity.

**Figure 3 jcm-11-06258-f003:**
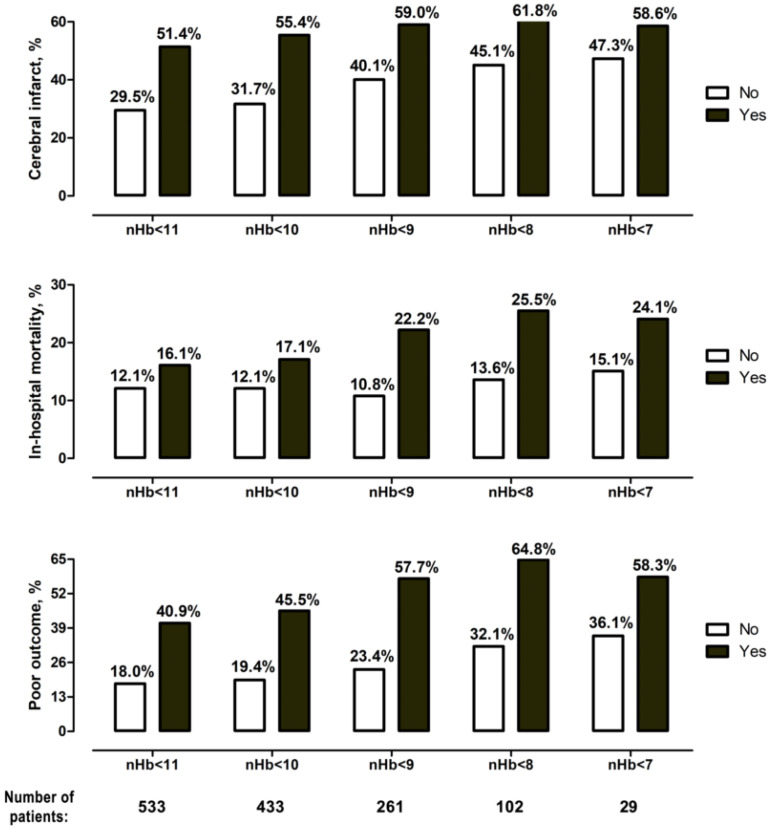
Occurrence of primary endpoints for the different degrees of anemia. There was a gradual increase in the rate of cerebral infarction, in-hospital mortality, and poor outcome for every nHb level decrease.

**Table 1 jcm-11-06258-t001:** Baseline characteristics of the final cohort.

Parameter	Number of Cases	Percentage *
Demographic characteristics and previous medical history		
Age ≥ 55	297	46.4%
Female sex	420	65.6%
Ethnicity (non-Caucasian)	28	4.4%
Hypertension	454	71.0%
Hypothyroidism	81	12.7%
Hyperthyroidism	5	0.8%
Hypercholesterolemia	53	8.4%
Hyperuricemia	18	2.9%
Diabetes mellitus Type II	38	6.0%
Statins	35	5.6%
Chronic painkiller abuse	47	7.5%
Blood thinners	56	8.8%
Initial characteristics of the SAH		
WFNS Grade (4–5)	283	44.2%
Fisher Grade (3–4)	532	88.8%
Acute hydrocephalus	488	76.3%
Aneurysm rebleed	31	4.8%
Clipping	262	40.9%
Aneurysm location ICA MCA AcoA ACA PC	67 152 230 22 169	10.4% 23.8% 35.9% 3.4% 26.4%
Aneurysm sack ≥ 7 mm	294	46.8%
Adverse events during the SAH		
Decompressive craniectomy	196	30.6%
Sonographic vasospasms	306	52.7%
Angiographically treated vasospasms	178	27.8%
Acute coronary syndrome	17	3.0%
Systemic infections	264	45.0%

Abbreviations: WFNS = World Federation of Neurosurgical Societies, ICA = internal carotid artery, MCA = middle cerebral artery, AcoA = anterior communicating artery, ACA = anterior cerebral artery, PC = posterior circulation. *—the percentages for each parameter were calculated according to the number of cases with known values.

**Table 2 jcm-11-06258-t002:** Univariate analyses of baseline characteristics as predictors of anemia in SAH.

	Nadir	< 7.0 g/dL	< 8.0 g/dL	< 9.0 g/dL	< 10.0 g/dL	< 11.0 g/dL
Hb Variable		OR (95% CI)				
**Demographic parameters and comorbidities**						
**Age ≥ 55 years**		1.95 (0.90–4.19)	**1.81 (1.18–2.79)**	**1.87 (1.36–2.57)**	**1.42 (1.01–1.98)**	1.36 (0.89–2.07)
Female sex		0.85 (0.40–1.84)	0.86 (0.56–1.34)	1.11 (0.80–1.55)	**1.54 (1.09–2.17)**	**2.36 (1.55–3.60)**
Ethnicity (non-Caucasian)		0.77 (0.10–5.89)	1.47 (0.58–3.71)	1.27 (0.60–2.72)	1.79 (0.72–4.49)	1.71 (0.51–5.76)
Hypertension		1.59 (0.64–3.98)	**1.82 (1.08–3.07)**	**1.45 (1.02–2.07)**	1.33 (0.93–1.91)	1.07 (0.68–1.69)
Hypercholesterolemia		1.27 (0.37–4.36)	1.59 (0.80–3.14)	1.34 (0.76–2.36)	**2.51 (1.20–5.24)**	2.03 (0.79–5.23)
Hypothyroidism		1.59 (0.59–4.29)	1.10 (0.58–2.09)	1.06 (0.66–1.73)	1.62 (0.93–2.83)	1.37 (0.68–2.76)
Hyperthyroidism		1.87 (0.10–34.63)	0.47 (0.03–8.63)	2.19 (0.36–13.21)	1.92 (0.21–17.29)	2.24 (0.12–40.79)
Hyperuricemia		2.71 (0.59–12.40)	2.05 (0.71–5.87)	1.87 (0.73–4.79)	1.26 (0.44–3.58)	1.64 (0.37–7.22)
Diabetes mellitus		1.18 (0.27–5.16)	1.20 (0.51–2.81)	1.49 (0.77–2.87)	**4.31 (1.51–12.33)**	**7.85 (1.07–57.84)**
Renal diseases		0.38 (0.02–6.47)	1.32 (0.48–3.58)	0.98 (0.43–2.21)	0.85 (0.37–1.96)	0.80 (0.29–2.18)
**Regular medication**						
Statins		1.38 (0.31–6.05)	1.68 (0.74–3.80)	1.24 (0.63–2.47)	**2.99 (1.14–7.83)**	2.17 (0.65–7.22)
Painkiller abuse		0.43 (0.06–3.24)	0.60 (0.23–1.57)	1.0 (0.55–1.83)	0.63 (0.34–1.15)	0.73 (0.35–1.52)
Blood thinners		2.28 (0.84–6.24)	1.49 (0.76–2.94)	1.39 (0.80–2.41)	1.84 (0.95–3.57)	1.75 (0.73–4.18)
**Initial SAH severity**						
WFNS grade (4–5)		2.14 (0.99–4.60)	**2.85 (1.83–4.45)**	**3.03 (2.19–4.21)**	**3.18 (2.21–4.57)**	**3.27 (2.01–5.31)**
Fisher grade (3–4)		7.91 (0.48–131.0)	**4.76 (1.46–15.46)**	**2.58 (1.44–4.64)**	**1.87 (1.12–3.15)**	**2.14 (1.17–3.91)**
Acute hydrocephalus		2.0 (0.68–5.83)	**3.30 (1.67–6.51)**	**3.02 (1.98–4.60)**	**2.83 (1.94–4.12)**	**2.75 (1.77–4.26)**
Aneurysm size ≥ 7 mm		1.54 (0.72–3.32)	1.22 (0.80–1.87)	1.15 (0.83–1.58)	1.14 (0.81–1.59)	1.24 (0.81–1.89)
**Clinical events and complications during the SAH**						
Aneurysm rebleed		2.3 (0.66–8.04)	1.54 (0.65–3.66)	1.31 (0.64–2.67)	2.15 (0.87–5.30)	1.99 (0.59–6.65)
Vasospasm		1.63 (0.75–3.51)	1.44 (0.92–2.27)	**1.74 (1.23–2.47)**	**2.78 (1.82–4.26)**	**3.58 (1.91–6.71)**
DC		1.64 (0.77–3.50)	**2.71 (1.76–4.18)**	**3.09 (2.18–4.37)**	**5.32 (3.31–8.55)**	**7.85 (3.57–17.24)**
ACS		1.36 (0.17–10.7)	3.0 (1.08–8.35)	2.14 (0.80–5.71)	2.33 (0.66–8.20)	1.46 (0.33–6.49)
Systemic infection		0.91 (0.42–1.97)	**1.71 (1.11–2.63)**	**2.28 (1.63–3.18)**	**3.23 (2.20–4.75)**	**3.92 (2.27–6.77)**
Clipping		0.64 (0.29–1.42)	**1.79 (1.17–2.73)**	**2.44 (1.76–3.37)**	**4.22 (2.86–6.24)**	**6.99 (3.75–13.05)**

Abbreviations: Hb = hemoglobin, OR = odds ratio, CI = confidence interval, WFNS = World Federation of Neurosurgical Societies, DC = decompressive craniectomy, ACS = acute coronary syndrome. Significant results are in bold.

**Table 3 jcm-11-06258-t003:** Overview of univariate and multivariate analyses for every nadir Hb level.

Nadir Hb:		< 7.0 g/dL	< 8.0 g/dL	< 9.0 g/dL	< 10.0 g/dL	< 11.0 g/dL
**Cerebral infarction**	UVA	1.58 (0.74–3.36)	**1.96 (1.27–3.03)**	**2.15 (1.56–2.97)**	**2.68 (1.89–3.80)**	**2.53 (1.61–3.97)**
	MVA M1	1.29 (0.58–2.87)	1.54 (0.96–2.46)	**1.72 (1.21–2.46)**	**2.47 (1.66–3.68)**	**2.30 (1.38–3.84)**
	MVA M2	1.28 (0.56–2.96)	1.33 (0.82–2.17)	**1.70 (1.16–2.50)**	**1.83 (1.19–2.81)**	1.67 (0.98–2.86)
**Unfavorable outcome at 6 months**	UVA	**2.48 (1.08–5.68)**	**3.88 (2.41–6.25)**	**4.47 (3.13–6.39)**	**3.47 (2.31–5.23)**	**3.15 (1.84–5.42)**
	MVA M1	1.98 (0.80–4.92)	**2.85 (1.65–4.92)**	**3.05 (2.04–4.56)**	**2.19 (1.39–3.46)**	**2.17 (1.18–3.98)**
	MVA M2	1.98 (0.74–5.34)	**2.21 (1.21–4.04)**	**2.44 (1.57–3.79)**	1.50 (0.89–2.52)	1.43 (0.75–2.72)
**In-hospital mortality**	UVA	1.80 (0.75–4.32)	**2.18 (1.31–3.63)**	**2.36 (1.52–3.64)**	1.5 (0.92–2.44)	1.39 (0.75–2.60)
	MVA M1	2.43 (0.64–9.21)	**3.32 (1.62–7.23)**	**4.84 (2.49–9.41)**	**9.61 (4.08–22.62)**	**7.41 (2.74–20.04)**
	MVA M2	2.20 (0.50–9.68)	**2.44 (1.07–5.55)**	**3.22 (1.59–6.51)**	**6.06 (2.39–15.34)**	**4.60 (1.61–13.14)**

Abbreviations: UVA = univariate analysis, MVA = multivariate analysis, M1 = model 1, M2 = model 2. Significant findings (*p* < 0.05) are in bold.

## Data Availability

Data are available and will be provided by the authors upon reasonable request.

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
