# Peer review of "Impact of Anemia Severity on the Outcome of an Aneurysmal Subarachnoid Hemorrhage"

_jcm, 2022, doi:10.3390/jcm11216258_

Round 1

Reviewer 1 Report (Previous Reviewer 1)

-

Author Response

-

Reviewer 2 Report (New Reviewer)

This study deals with relationship between nadior hemoglobin and functional outcomes of SAH patients.

The author has submitted quite a fascinating manuscript. I have found a few issues that, once addressed, will improve the manuscript.

The  "Table 3" and "Figure 3" are very important. They show a gradual increase according to the degree of anemia except "nHB<7". The author should discuss the reason of exception for "nHB<7".

In this article the author described only nHB <11, he should deal with all nHB with or without anemia, in all ranges.

In this study, comparison of nHB with before onset is necessary.

Author Response

This manuscript is a resubmission of an earlier submission. The following is a list of the peer review reports and author responses from that submission.

Round 1

Reviewer 1 Report

The authors present an interesting manuscript focusing on a highly important topic. The manuscript is well written.

However, there is few remarks and comments.

1) The authors may consider to add number of patients to each endpoint and columns in the Figure 3.

2) Did the authors take into account the level of intravenous hydration for each patient and its impact on the level of anemia? Heavy intravenous hydration may also decrease hemoglobin physiologically.

3) Did all patients in the study cohort receive RBCT who had a decrease in nadir Hb < 7.0g/dL? In addition, did some other patients in the study cohort receive RBCT?

Reviewer 2 Report

Observational studies are important because they identify mechanisms and provide impetus to future research(1). At present, rapid advances in neurosurgical knowledge and technology have been observed(2), but rupture of aneurysms still leads to a devastating condition(3) with its unpredictable behavior and its dismal prognosis(4). For that reason, the outcome of patients with SAH is an important issue. The authors investigated the outcome-relevant severity of post-SAH anemia. The subject of the manuscript is not new, because it is known that a negative impact of anemia on the outcome of aneurysmal subarachnoid hemorrhage (SAH).  As the primary study endpoints, the authors investigated the impact of post-SAH anemia severity on the occurrence of the following major outcome events: 1) occurrence of cerebral infarction(s); 2) in-hospital mortality; 3) unfavorable outcome at 6 months post-SAH. There are many studies on this subject. What is the difference of the present study from previous studies? What does this study a new data to current medical literature? The authors discussed this point, but these points should be briefly mentioned in the conclusion section

References

1.        Polat HB, Kanat A, Celiker FB, Tufekci A, Beyazal M, Ardic G, et al. Rationalization of using the MR diffusion imaging in B12 deficiency. Ann Indian Acad Neurol. 2020;23(1):72–7.

2.        Kanat A, Yazar U, Kazdal H, Yilmaz A, Musluman M. Neurosurgery is a profession. Neurol Neurochir Pol. 2009;43(3):286–8.

3.        Aydin MD, Kanat A, Yolas C, Soyalp C, Onen MR, Yilmaz I, et al. Spinal subarachnoid hemorrhage induced intractable miotic pupil. A reminder of ciliospinal sympathetic center ischemia based miosis: An experimental study. Turk Neurosurg. 2019;29(3):434–9.

4.        Aydin MD, Kanat A, Sahin B, Sahin MH, Ergene S, Demirtas R. New Experimental Finding of Dangerous Autonomic Ganglia Changes in Cardiac Injury  following Subarachnoid Hemorrhage; A reciprocal culprit-victim relationship between the brain and heart. Int J Neurosci. 2022 Jun;1–15.

Round 2

Reviewer 2 Report

The revision was not well done. The manuscript does not a new data  to current literature